# Establishing Experimental Conditions to Produce Lignin-Degrading Enzymes on Wheat Bran by *Trametes versicolor* CM13 Using Solid State Fermentation

Paul W. Baker *  and Adam Charlton

Biocomposites Centre, Bangor University, Deiniol Road, Bangor LL57 2UW, Wales, UK; adam.charlton@bangor.ac.uk
* Correspondence: paul.baker@bangor.ac.uk; Tel.: +44-(0)-124-838-2640

**Abstract:** Valorisation of wheat bran can be achieved by solid state fermentation (SSF), through application of this material as a growth substrate for a natural white rot fungal isolate, *Trametes versicolor* CM13, to produce lignin-degrading enzymes. One of the main challenges in optimising and upscaling (SSF) processes is the accurate adjustment and maintenance of moisture conditions. This factor was assessed in the scale up of microcosms and was evaluated over 28 days, under two slightly different moisture contents, reflecting minor differences in experimental conditions during set up and operation of the SSF process. In addition, the microcosms were processed differently from the initial trial using homogenisation of whole microcosms to create a homogeneous mixture prior to sampling. This appeared to result in less variation among the collected samples from the microcosms. Variation of measured parameters as a percentage of actual values measured ranged from 1.33% to 144% in the unmixed microcosms and from 0.77% to 36.0% in the pre-mixed microcosms. Decomposition in the more saturated microcosms progressed more quickly as hemicellulose content decreased and reached a steady state after 14 days, whereas hemicellulose content continued to decrease until 21 days in the less saturated microcosms. Lignin-degrading enzyme activities were not significantly different between either sets of experiments except for laccase on day 7. Laccase and manganese peroxidase activities were highest on day 21 and were similar in both sets of experiments. Enzyme activities on day 21 in the microcosms at moisture content of 42.9% and at 54.6% for laccase activities were $750 \pm 30.5$ and $820 \pm 30.8$ units, and for manganese peroxidase, activities were $23.3 \pm 6.45$ and $21.4 \pm 21.4$ units, respectively. These results revealed different decomposition rates during the early stage of solid-state fermentation as a function of the initial moisture content, whereas final enzyme activities and fibre content during the later stage were similar in microcosms having different moisture contents at the start.

**Keywords:** fungi; laccase; microcosm; homogenisation; hemicellulose; cellulose; fibre; biomass



## 1. Introduction

One potential application for agricultural crop waste is as a substrate for solid-state fermentation (SSF) with fungi, to produce industrial enzymes that have increasing uses and demand, particularly in the food industry. In terms of total global tonnage produced, wheat, rice and corn are the predominant crops globally, leading to large quantities of waste by-products [1]. Wheat is the major crop produced in Europe, and wheat bran is one of the by-products constituting 14–19% of the whole grain. Europe is estimated to produce 100 million tonnes of wheat bran annually [2] and some of this wheat bran could be used as a substrate for enzyme production using SSF with higher valorisation potential. SSF involves the growth of microorganisms, particularly fungi, on a moist, solid substrate but in the absence of free-flowing water [3]. The advantage of SSF, compared with the current, commercial processes for manufacturing enzymes using liquid (submerged) fermentation,

is the reduced water input and the increased product output [4]. Submerged fermentation processes can give rise to dilute product streams and the removal of water is a costly process step that can be avoided using SSF [4]. Concentrated enzymes extracts were recovered from SSF during downstream processing with additional water. The majority of fungal SSF processes have focused on use of the Ascomycetes, particularly the *Aspergillus* and *Trichoderma* species, and while some studies described using Agaricomycetes, most of these were *Pleurotus* species [5]. Many of these studies may have used *Pleurotus* sp. because this fungus readily colonises and degrades a wide range of lignocellulose substrates. Consequently, another potential advantage of SSF when using these fungi is that the residual lignocellulosic material, after enzyme extraction, has potential application in downstream fermentation of the accessible sugars to a range of products, including biofuels and platform chemicals. In this study, *Trametes versicolor* CM13, a white rot fungus, and a natural isolate from an oak log, was used as an alternative to other fungi previously investigated, because it has been reported to exhibit high enzyme activity linked to the degradation of different wood species [6].

White rot fungi produce laccase, manganese peroxidase, lignin peroxidase and versatile peroxidase, which are the main groups of extracellular, lignin-degrading enzymes previously investigated, although some bacteria, e.g., *Pseudomonas* sp. and *Cupriavidus basilensis,* also possess peroxidases and laccases, resulting in the production of lipids and polyhydroxyalkanoate [7]. Lignin is a major component within woods and grasses and is a highly crosslinked, large molecular weight polymer, composed of three different monolignols (*p*-coumaryl alcohol, guaiacyl alcohol and sinapyl alcohol) and polyphenolic compounds [8]. The combinations of the lignin-degrading genes can vary significantly between different fungal species where, for example, *T. versicolor* was shown to possess 12 short manganese peroxidase genes and 10 lignin peroxidase genes in the whole genome [9]. A previous study reported that manganese peroxidase activity was higher compared with lignin peroxidase activity, which was often barely detectable [10]. Purified manganese peroxidase from *Ceriporiopsis subvermispora* showed significant degradation of a radioactively labelled lignin construct, linked to a polyethylene glycol backbone, and degradation increased as function of increasing enzyme concentration [11]. The role of manganese peroxidase in this process was further substantiated with the use of a recombinant manganese peroxidase, which revealed chemical changes in the aliphatic chains connected to the aromatic rings. In contrast, another study reported that the aromatic components within the lignin macromolecule were mineralised by lignin peroxidase [12]. Later studies have shown that higher levels of lignin degradation were achieved using a combination of both laccase and manganese peroxidase [13,14]. Consequently, manganese peroxidase has potential industrial application in degrading textile dyes [15] and lignin degradation for biofuel production in the pulp and paper industry [16,17].

One of the potential problems associated with SSF is the accurate adjustment of the moisture content at the start and minor changes in moisture content during fermentation that deviate from the optimal conditions for rapid fungal growth. Most lignocellulosic substrates have a heterogenous composition in terms of cell types and sizes that will affect water uptake. Furthermore, maintaining the same moisture content is impossible when using forced aeration because this caused evaporation, while, conversely, biomass decomposition leads to increased saturation. Consequently, moisture content has been highlighted as one of the most important parameters affecting the efficiency and potential upscaling of fungal SSF processes [18]. Therefore, the aim of this study was to examine the effect of minor differences in the initial moisture content on the outcome of the lignin-degrading enzymes, along with fibre decomposition.

## 2. Materials and Methods

### 2.1. Particle Analysis of Wheat Bran

Triplicate samples of wheat bran (20 g) were separated through overlapping 3 cm diameter sieves on the Octogon 200 sieve shaker (Endecotts, London, UK) for 10 min at an

amplitude of 2.7 mm at 3000 min$^{-1}$ and 50 Hz. The material collected in each fraction was weighed.

## 2.2. Analysis of Growth of T. versicolor CM13 on Wheat Bran over Time

Microcosms containing wheat bran (20 g) were adjusted with deionised water (15 mL), autoclaved, and then inoculated with spores collected from *T. versicolor* CM13 growing on malt extract agar for one month. Sterile deionised water (10 mL) was added to the plates and the surface was scrapped aseptically with a loop. The spore suspension was pipetted from the plates into 30 mL sterile deionised water and mixed thoroughly. Microscopy revealed a spore population of about $1 \times 10^7$ cells per ml with only a few fragments of aerial mycelium being present. This suspension (3 mL) was added to each microcosm with mixing so that the final moisture content was 47.4%. All microcosms were incubated at 22 °C and when initial growth was observed, duplicate microcosms were destructively sampled by mixing the entire contents with a spatula and collecting two samples (5 g). One of these samples was used for moisture analysis and the resulting dried material was then used for fibre analysis, while the other sample was used for enzyme extraction and pH measurement.

## 2.3. Effect of Enzyme Activities of T. versicolor CM13 at Two Different Moisture Contents

The experiment was repeated as described in the previous section, except that a larger quantity of material was used to reduce differences caused by rates of evaporation. Two sets of microcosms were prepared with wheat bran (100 g) containing deionised water, one with a final moisture content of 42.9% and another with 54.6%. The moisture content at 42.9% represented the lowest possible moisture content where fungal growth occurred and the moisture content at 54.6% was sufficiently high enough to be statistically different from the moisture content at 42.9% at $p < 0.05$. Spores were collected from the surface of *T. versicolor* CM13 growing on malt extract agar for one month in sterile distilled water, and the spores were inoculated into the microcosms with mixing. The microcosms were destructively sampled in triplicate on days 7, 14 and 21, as described in the previous section, by mixing the degraded substrate with a sterile spatula. The entire microcosms were then homogenised (Waring blender BB255SK, Conair Corporation, Stamford, CT, USA) for 60–150 s until the clumps of mycelia were no longer visible. Samples (5 g) were removed from each microcosm for moisture analysis, fibre analysis, extraction of enzymes and pH measurements.

## 2.4. Analysis of Samples

On the sampling day, the contents of the microcosm were mixed using a spatula and 5 g was removed to determine moisture analysis, which was dried at 103 °C until the rate of moisture loss was less than 20 mg/min. The dried material ($0.5 \pm 0.025$ g) was also used in stepwise fibre analysis using neutral detergent fibre (NDF) and acid detergent fibre (ADF) solutions in the Fibre analyser (Ankom, New York, NY, USA) machine to determine non-fibre, hemicellulose and cellulose contents, using a reported method [13]. The insoluble lignin content in the bags was determined after ADF extraction using 72% (*v/v*) sulphuric acid (500 mL) in the Daisy machine (Ankom, New York, NY, USA) by rotating at room temperature (3 h). The bags were washed repeatedly in 2 L tap water until pH 7. Ash contents were determined by heating 0.5 g of the original material before fibre analysis in crucibles in a furnace (600 °C for 4 h). The calculated ash contents in each fibre were subtracted from each fibre fraction assuming equal distribution of ash among the different types of fibre.

Klason lignin contents were determined on a dried sample (0.3 g) using 72% sulphuric acid (5 mL), which was incubated at 25 °C for 3 h and was regularly stirred. Deionised water (240 mL) was added to the suspension, which was then autoclaved (121 °C, for 1 h). The hot suspensions were filtered through pre-weighed glass microfibre filters (Whatman, Little Chalfont, Buckinghamshire, UK) along with additional hot deionised water to remove

any remaining soluble compounds. The filters were oven dried (103 °C) overnight and weighed and the lignin content was determined by subtracting the weight of the filter. The ash contents were subtracted from these values.

Another sample (5 g) was resuspended in deionised water (100 mL), blended (1 L Waring blender) and filtered through cotton wool. The pH in this filtrate was measured. A sample was removed from the crude filtrate and filtered through a 0.2 μm polycarbonate filter to remove any remaining solids that could increase the background colour of the colorimetric assays.

Laccase activity was determined using 400 μL 3 mM ABTS (2,2′-azino-bis (3-ethylbenzo thiazoline-6-sulfonic acid), 400 μL 1 mM sodium acetate buffer, pH 4.5 and 400 μL of sample and the absorbance values of the samples were measured at 420 nm. The control sample consisted of 400 μL deionised water instead of the sample. Laccase activity was calculated based on the molar extinction coefficient of ABTS being $\varepsilon_{420}$, 36,000 $M^{-1} \cdot cm^{-1}$ [19].

Manganese peroxide was determined in 1.1 mL of 50 mM sodium succinate (pH 4.5), 50 mM sodium lactate (pH 4.5) using 0.1 mM $MnSO_4$, 0.1 mM phenol red with 50 μL of sample [20]. Activity was initiated with 50 μM hydrogen peroxide (prepared as 20 μL of 0.43%) and at the end of 30 min incubation, 57.5 μL of 10% sodium hydroxide was added. The absorbance values of the samples were measured at 610 nm and calculated using the molar extinction coefficient ($\varepsilon_{610}$, 30,737 $M^{-1} \cdot cm^{-1}$). An experimental control was performed using deionised water instead of the sample.

Another assay to measure decolourisation activity was performed with Remazol brilliant blue (800 μL, 0.16 mM) in water with a 400 μL sample and hydrogen peroxide (20 μL, 38 mM). Decolourisation of this azo dye is indicative of the combined activities of laccase and manganese peroxidase, where one or the other enzyme compensates for the other enzyme when the activity of one enzyme is higher than the other [21]. The absorbance values of the samples were measured at 592 nm and activity was calculated based on the molar extinction coefficient of Remazol brilliant blue being $\varepsilon_{592}$, 6170 $M^{-1} \cdot cm^{-1}$ [22]. The sample changed the absorbance value of the assay and, consequently, it was necessary to include the sample within the control. Experimental controls were performed using 400 μL of the samples and deionised water (20 μL), instead of hydrogen peroxide.

Lignin peroxidase activity was determined in 125 mM sodium tartrate (pH 3) using 0.16 mM azure B and 5 mM hydrogen peroxidase based on a similar method previously described [23]. The absorbance values of the samples were measured in a microplate reader at 651 nm ($\varepsilon_{651}$, 48,800 $M^{-1} \cdot cm^{-1}$). The same method of experimental controls was performed as with Remazol brilliant blue using the sample instead of water.

All enzyme activities were determined using pre-warmed reagents incubated (30 °C) with horizontal shaking (600 rpm). At the end of the incubation (30 min), 100 μL of each sample was measured in a microplate reader (BioTek Epoch, Agilent, Santa Clara, CA, USA) at the respective absorbance value. Enzyme activities were calculated using the molar extinction coefficient of the coloured compound being measured. Enzyme units are μM of the coloured compound formed or decolourised per min. The pH values were measured in the remaining filtrates.

### 2.5. Statistical Analysis

Growth studies of *T. versicolor* CM13 on wheat bran over time and under different moisture conditions were conducted using duplicate and triplicate microcosms, respectively, and single samples were analysed from each microcosm. Data analysis was conducted using ANOVA (analysis of variance) in SPSS version 27 with Tukey's post hoc test. The averages and standard deviations are shown in the figures and tables, while superscript letters represent significant differences. Those with the same letters are similar and those with different letters are significantly different at $p < 0.05$.

## 3. Results and Discussion

### 3.1. Growth of T. versicolor on Wheat Bran over Time

In relation to the size distribution of the wheat bran, it was determined that 91% of particles ranged from 0.5 to 2 mm (Figure 1), and this can have a significant impact on the efficiency of SSF processes due to increased uptake of water by the growth substrate as a function of reducing particle size. For example, the thermo-mechanical pre-treatment of *Miscanthus*, reported previously, using continuous pressurised disc refining resulted in the formation of smaller sized particles (<250 μm) compared to the original chopped material (>1 mm), and lower moisture contents were required to achieve saturation in the refined material [11]. The potential benefits of lower moisture contents could lead to increased gaseous exchange and more rapid dissipation of heat, which are critical factors effecting SSF efficiency. Wheat bran is composed of three different tissues, the outer pericarp, an intermediate strip and the aleurone layer possessing elastic properties [24]. In addition, a minor proportion of the seed starch (~19%) becomes associated with wheat bran due to the imperfect detachment of the bran layer from the grain [2]. Consequently, it assumed that any moisture is either directly absorbed due to the water binding capacity of the fibres or forms a thin film around the wheat bran particles, which may limit the amount of water absorbed by the bran. This is supported by previous studies which reported that the moisture content for the optimal growth on wheat bran of *Trichoderma reesei* QM9414 was ~46% [25] and for generating high cellulase activity from a mutagenic strain of *Aspergillus* sp. was 33% [26]. These values are considerably lower than the atypical low moisture content of 67%, which only supported the growth of only one species of white rot fungi, *Pleurotus erygnii*, on wheat straw [27]. This study also reported that the addition of 30% wheat bran to wheat straw reduced the moisture content to 44%, but this level of moisture continued to support the growth of *P. erygnii*.

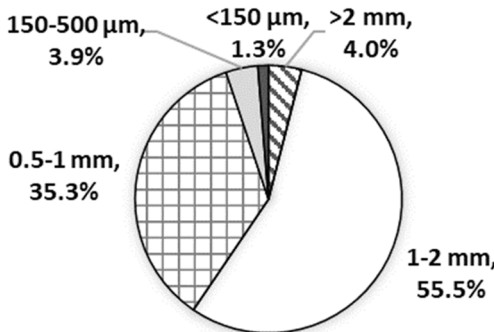

**Figure 1.** Percentage distribution of particle sizes of dry wheat bran before inoculation with *T. versicolor* CM13 spores which was determined using a sieve shaker.

The percentage distribution of hemicellulose, cellulose and lignin in wheat bran were 22.7%, 6.8% and 3.2%, respectively, with the remaining 67.3% composed of the non-fibre component. These values compare well with previously reported values [28]. During SSF, fungal degradation caused the non-fibre content to decrease on days 25 to 37 compared with days 0 to 13, which was statistically significant (Figure 2), possibly indicating that the presence of starch [2], pectins and soluble acetylated hemicelluloses [29] that contributed to fungal growth. Arabinoxylans form about 50% of the dry weight present in wheat bran [21], which is higher than the total fibre content, indicating that a significant proportion of these are soluble. In addition, the hemicellulose content showed a similar decrease over time and was significantly lower on days 13 and days 25 to 37 compared with day 7, whilst the cellulose content significantly increased on days 25 to 37 compared with days 0 to 13. These results indicate that the fungus was producing hemicellulases rather than cellulases, which was indicated by the decreasing quantities of hemicellulose lost through decomposition whereas cellulose content increased. Furthermore, the quantities of hemicellulose in wheat bran were higher than cellulose that may facilitate the fungus to produce hemicellulases.

Similar patterns have been observed with mixed substrates of wheat straw and wheat bran appearing to suppress cellulose degradation [27,30]. However, fungal degradation of wheat straw without wheat bran showed significant cellulose decomposition. Similarly, the insoluble lignin content was significantly higher on days 13 and 37 compared with days 0 and 7. Consequently, the lignin content showed a decrease only during the initial stage of decomposition. Lignin is closely associated with hemicellulose and cellulose, and, presumably, the degradation of lignin with increasing lignin-degrading enzyme activity is accompanied with increased hemicellulose decomposition. A previous study highlighted that the addition of increasing quantities of wheat bran (2% to 10%) to wheat straw had no effect on delignification, possibly due to the higher nitrogen content associated with wheat bran which might be expected to lower the rate of delignification [31]. The ash content appeared to increase with incubation time and was higher on day 33 compared with days 0–18, indicating a loss in biomass. The increase in ash content resulted from decomposition of organic material, rather than inorganic components which presumably remain intact during fungal growth.

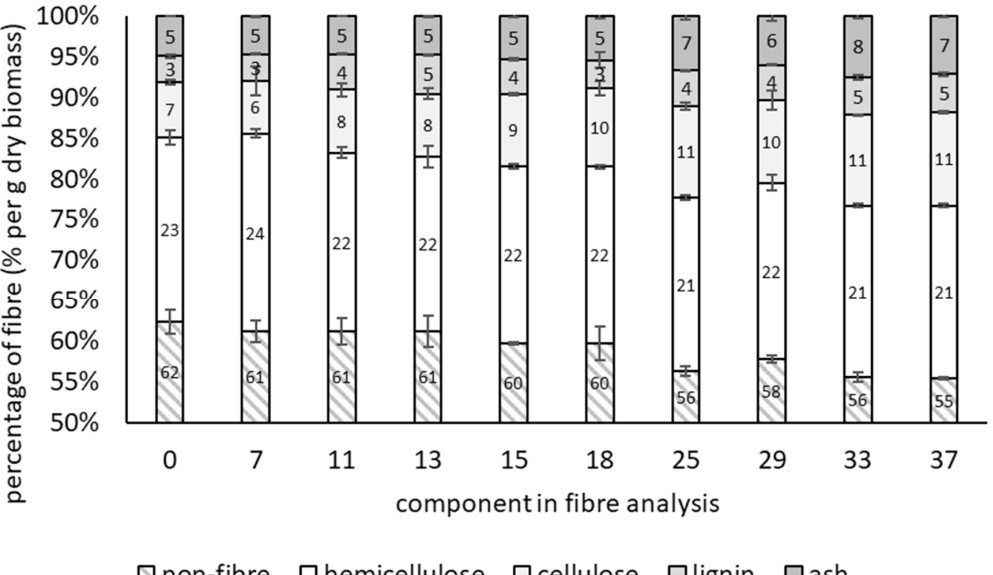

**Figure 2.** Bars represent the proportions of fibre content along with standard deviations during fungal degradation by *T. versicolor* growing on wheat bran. The decrease in non-fibre content may be due to soluble hemicelluloses acting as nutrients for the fungus.

During the SSF experiments, the pH values decreased over time and were lowest between days 11 and 18 (Figure 3), and this was probably due to organic acid production which was observed by some strains of filamentous fungi [32]. The moisture contents, manganese peroxidase activities and Klason lignin contents (Figures 3 and 4) showed no significant differences over time. These results showed that moisture content was kept constant for the duration of the experiment. However, it was expected that manganese peroxidase activities and Klason lignin content would have changed over the experiment. This is because there appeared to be an increase in manganese peroxidase activities on day 25 and Klason lignin contents appeared to decrease on day 15, which is linked to peak laccase activities. However, the lack of significant differences may be due to the high errors associated with some of the data. Klason lignin values were relatively high, and this may be due to the presence of cutin [33]. Laccase activity was significantly higher on days 13 to 33 compared with days 7, 11 and 37. Laccase activities appeared to increase and decrease over this period perhaps reflecting the expression of redundant genes at different stages during the growth cycle [34]. Laccase activity initially peaked on day 15 and there were two phases of manganese peroxidase activity which peaked on days 11

and 25, while. A similar trend was observed with the growth of *Inonotus obliquus* on wheat bran, although higher manganese peroxidase activity was determined during the first phase [35]. A previous study has shown that the occurrence and length of expression of lignin-degrading enzymes was dissimilar among different species of white rot fungi, when growing on wheat straw [10].

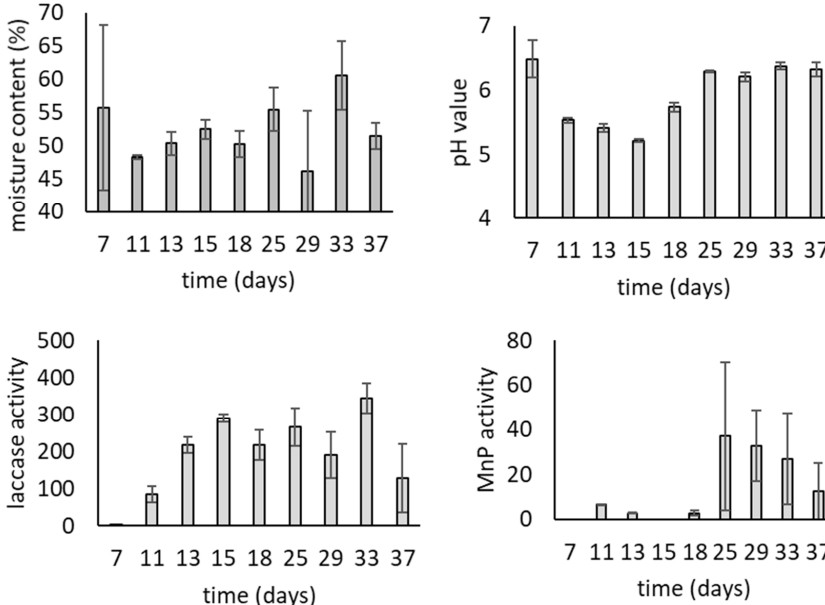

**Figure 3.** The moisture contents of duplicate microcosms destructively sampled after initial colonisation of wheat bran by *T. versicolor* CM13. The changes in pH possibly relate to the production of organic acids formed by this fungus. Laccase activities increased early and appeared to fluctuate during the experiment whereas manganese peroxidase activities (MnP) were mostly determined later in the experiment.

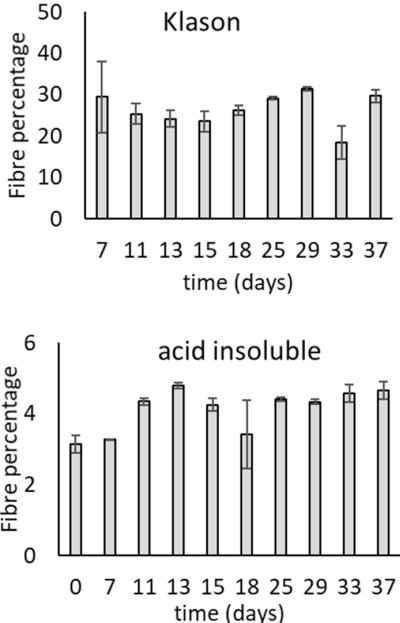

**Figure 4.** Quantities of Klason lignin (total lignin) and acid insoluble lignin (after removal of hemicellulose and cellulose) in wheat bran degraded by *T. versicolor* CM13.

### 3.2. Growth of T. versicolor CM13 on Wheat Bran under Two Different Moisture Conditions

Visual examination of the microcosms clearly showed complete colonisation of the microcosms, but some regions showed extensive mycelial growth often near the glass vessel surface. A previous study using scanning electron microscopy reported a network of fungal mycelial clumps (9–12 μm diameter) of *T. versicolor* within a network of growing mycelia on semi-solid medium containing suspended wheat bran [36]. Consequently, samples randomly removed from the microcosms may differ significantly in fungal biomass contents due to this irregular growth of mycelia. Furthermore, it was evident that these mycelial clumps often remained intact during the homogenisation extraction step to recover the enzymes and, therefore, another homogenisation step was incorporated beforehand to process the whole microcosm rather than a small sub-sample.

The use of homogenisation (high shear mixing) to redistribute the fungal mycelium more evenly in the whole microcosm, resulted in lower standard deviations (Table 1), following measurements from a sample of this material. Standard deviations associated with the smaller microcosms (20 g) were greater where sampling was performed without prior homogenization. The effects of evaporation were less pronounced in the larger microcosms due to the larger quantity of biomass.

**Table 1.** Standard deviations of SSF experimental parameters in the microcosms as a proportion of actual measured values (represented as percentages).

| Measured Parameter | Time | Saturation | |
|---|---|---|---|
| Microcosm size | 20 g | 20 g | 100 g |
| Prior homogenisation | no | no | yes |
| Moisture content | 7.90 | 15.1 | 2.25 |
| Biomass loss | ND | 19.8 | 7.53 |
| pH | 1.33 | 3.37 | 0.77 |
| Laccase | 32.7 | 36.1 | 6.13 |
| Manganese peroxidase | 34.7 | 98.4 | 36.0 |
| Remazol Brilliant Blue activity | ND | 144 | 14.8 |

The moisture contents between the microcosms at 42.9% and at 54.6% were sufficiently close to each other to mimic minor variations around a midpoint of 48.8% that could occur when establishing the microcosms and during operation yet were also significantly different at the start of the experiment (Table 2). The moisture contents of both sets of microcosms at the start of the experiment were within the range for optimal cellulase production by *Aspergillus* sp., growing on wheat and reported in a previous study [26]. Both sets of microcosms differed from each other by 11.8% and both showed a significant increase in moisture content over time, which corresponded with significant biomass decomposition. The final moisture contents of both sets of microcosms differed from each other by 6.1% at the end of the experiment but these results were not significantly different, unlike the moisture contents of both sets of microcosms at the start. Therefore, moisture contents appeared to converge over time during fungal decomposition. However, these microcosms received passive aeration and the effect of forced aeration would undoubtedly have resulted in a different outcome. The moisture content in the more saturated microcosms appeared to remain unchanged from day 14 onwards and this was reflected by the slow increase in biomass loss, pH changes, and laccase and lignin-degrading activities. This indicated that rapid fungal colonisation occurred within 14 days and, thereafter, entered a stationary phase where there was limited growth. In contrast, the moisture content of the less saturated microcosms showed a slower rate of increase that continued until day 21, at which point the moisture contents of both sets of microcosms were not significantly different. Fungal decomposition as shown by biomass loss, pH changes and hemicellulose decomposition revealed that the pattern was similar in the less saturated microcosms as in the more saturated microcosms except there was a lag phase of 7 days in the less saturated microcosms. A slower rate of decomposition occurred from day 14 in the more

saturated microcosms that was indicative of stationary phase growth. It was apparent that the stationary phase was achieved on day 21 in the less saturated microcosms. Fungal decomposition during the stationary phase progresses more slowly and, therefore, any differences between the microcosms will become minimal.

**Table 2.** Physical parameters and lignin enzyme activities of wheat bran colonised by *T. versicolor* CM13 at different moisture contents.

| | Day | Low | High |
|---|---|---|---|
| Moisture content | 0 | 42.9 ± 0.18 [a] | 54.6 ± 0.80 [b] |
| | 7 | 54.4 ± 3.87 [b] | 57.0 ± 1.73 [bc] |
| | 14 | 58.9 ± 1.61 [b] | 67.5 ± 1.61 [d] |
| | 21 | 62.6 ± 1.40 [cd] | 67.7 ± 0.95 [d] |
| Biomass loss | 7 | 27.7 ± 6.06 [a] | 32.9 ± 5.28 [a] |
| | 14 | 26.5 ± 2.79 [a] | 43.7 ± 3.29 [b] |
| | 21 | 45.1 ± 0.83 [b] | 46.7 ± 0.81 [b] |
| pH | 0 | 6.22 ± 0.03 [ab] | 6.25 ± 0.01 [b] |
| | 7 | 5.55 ± 0.11 [a] | 5.96 ± 0.09 [b] |
| | 14 | 5.80 ± 0.04 [c] | 6.10 ± 0.01 [bc] |
| | 21 | 6.22 ± 0.02 [ab] | 6.27 ± 0.04 [a] |
| Laccase | 0 | 0.12 ± 0.00 [a] | 0.58 ± 0.59 [a] |
| | 7 | 366 ± 25.6 [b] | 628 ± 27.7 [c] |
| | 14 | 529 ± 87.5 [c] | 777 ± 25.2 [d] |
| | 21 | 750 ± 30.5 [d] | 820 ± 30.8 [d] |
| Manganese peroxidase | 7 | 1.69 ± 0.23 [a] | 1.23 ± 0.83 [a] |
| | 14 | 3.41 ± 1.47 [a] | 6.85 ± 3.46 [a] |
| | 21 | 23.3 ± 6.45 [b] | 21.4 ± 2.54 [b] |
| Lignin peroxidase | 7 | ND | ND |
| | 14 | 0.47 ± 0.29 [a] | 0.55 ± 0.36 [ab] |
| | 21 | 0.00 ± 0.66 [ab] | 0.00 ± 4.16 [ab] |
| Lignin-degrading activity | 7 | 0.00 ± 54.66 [a] | 107 ± 114 [a] |
| | 14 | 632 ± 14.5 [b] | 792 ± 43.7 [b] |
| | 21 | 72.4 ± 39.1 [b] | 102 ± 10.1 [b] |

Numbers with the same letters denotes similarity and those with different letters are significantly different at $p < 0.5$.

In general, the manganese peroxidase activities showed a significant rate of increase that were not significantly different from each other at the end of the experiment, but both had significantly increased compared with activities measured on days 7 and 14. In another study, a comparison was made in the growth of *Pleurotus pulmonarius* on wheat bran at two different moisture contents, which revealed higher manganese peroxidase activity at a moisture content of 75% and higher laccase activities at a moisture content of 91% [37]. However, our study indicates that neither laccase activities nor manganese peroxidase activities showed any difference between the two sets of microcosms, although the moisture conditions in our study were 30–43% lower. Lignin peroxidase activities were low, and the highest activities were obtained on day 14, under both sets of moisture conditions.

Lignin-degrading activities, represented by the decolourisation of Remazol brilliant blue, exhibited significant increases in both set of microcosms on days 14 and 21. Activities were higher on day 14 in both sets of microcosms, but these values were not significantly different to the lower activities determined on day 21. These enzyme activities were similar even though hemicellulose decomposition was different during this period. Previous evidence indicates that Remazol brilliant blue can be degraded by laccase [38,39], manganese peroxidase [40–42] and lignin peroxidase [43]. Consequently, it is possibly indicative of overall delignifying enzyme activity, although another study reported that in studies using purified enzymes, only laccase, and not manganese peroxidase, was effective [44]. This may indicate that only certain manganese peroxidases can degrade Remazol brilliant blue. Our study appears to indicate that laccase, manganese peroxidase and lignin per-

oxidase were all involved in degrading Remazol brilliant blue, because all these enzymes showed activity during day 14, but none exhibited peak activities that was indicative of single-enzyme-mediated degradation.

The remaining hemicellulose contents in the less saturated microcosms showed a significant decrease after 7 days incubation, which was more noticeable in the more saturated microcosms, but, thereafter, showed only minor decreases in both sets of microcosms (Table 3). The conditions in the more saturated microcosms were more favourable for rapid fungal growth, as evidenced by the greater reduction in the remaining hemicellulose content. Although the quantity of hemicellulose in the less saturated microcosms appeared higher at the end of the experiment than in the more saturated microcosms, this was not significantly different. The cellulose contents in both sets of microcosms showed a similar increase and neither showed significant differences between each of the microcosms on the respective days. These results correlate well with another study on wheat bran, which used NMR analysis to determine the level of hemicellulose degradation by *Pleurotus ostreatus* after 62 days incubation, resulting in a total dry weight loss of 20% [45]. The fibre composition of the two sets of microcosms were highly similar when complete colonisation had occurred on day 21 even though growth patterns during the initial stages were dissimilar. Decomposition occurred more rapidly in the more saturated microcosms which was followed by a slower decomposition rate whereas decomposition was delayed in the less saturated microcosms and continued until day 21.

**Table 3.** Effect of moisture content of fibre degradation (white area) and percentage of total fibre degraded (light grey area) by *T. versicolor* CM13.

| Moisture Content | Time (Days) | Non-Fibre | Hemicellulose | Cellulose | Lignin | Ash |
|---|---|---|---|---|---|---|
| 43.9%, 54.6% | 0 | 62.4 ± 1.53 [a] | 22.7 ± 0.92 [a] | 6.82 ± 0.31 [a] | 3.23 ± 0.24 [a] | 5.01 ± 0.44 [a] |
| 43.9% | 7 | 60.1 ± 1.23 [a] | 21.7 ± 0.79 [b] | 9.62 ± 0.45 [ab] | 3.84 ± 0.05 [a] | 3.28 ± 0.13 [b] |
| | 14 | 60.1 ± 0.58 [a] | 20.5 ± 0.52 [bc] | 10.7 ± 0.49 [ab] | 3.79 ± 0.23 [a] | 3.51 ± 0.08 [b] |
| | 21 | 58.0 ± 0.69 [a] | 21.1 ± 0.06 [bc] | 11.5 ± 0.61 [b] | 4.31 ± 0.06 [a] | 3.61 ± 0.51 [b] |
| 54.6% | 7 | 61.8 ± 0.54 [a] | 19.5 ± 0.39 [c] | 9.81 ± 0.24 [ab] | 3.65 ± 0.13 [a] | 3.65 ± 0.66 [b] |
| | 14 | 62.6 ± 5.09 [a] | 20.7 ± 1.38 [bc] | 9.73 ± 3.43 [ab] | 1.96 ± 3.04 [a] | 3.53 ± 0.23 [b] |
| | 21 | 58.2 ± 0.74 [a] | 19.4 ± 0.51 [c] | 12.5 ± 0.42 [b] | 4.46 ± 0.17 [a] | 4.00 ± 0.17 [ab] |
| 43.9% | 7 | 12.29 ± 4.39 [a] | 6.63 ± 0.80 [a] | 0.17 ± 0.55 [a] | 0.22 ± 0.20 [a] | |
| | 14 | 15.46 ± 2.79 [ab] | 8.55 ± 1.40 [ab] | 0.00 ± 0.80 [a] | 0.47 ± 0.16 [a] | |
| | 21 | 23.95 ± 0.85 [c] | 10.72 ± 0.15 [c] | 0.80 ± 0.24 [a] | 0.64 ± 0.05 [a] | |
| 54.6% | 7 | 10.32 ± 1.98 [a] | 7.95 ± 0.28 [a] | 0.00 ± 0.37 [a] | 0.26 ± 0.13 [a] | |
| | 14 | 19.69 ± 1.12 [bc] | 10.36 ± 0.19 [bc] | 1.45 ± 2.26 [a] | 0.54 ± 0.20 [a] | |
| | 21 | 24.76 ± 0.87 [c] | 11.96 ± 0.16 [c] | 0.46 ± 0.20 [a] | 0.62 ± 0.07 [a] | |

Numbers with the same letters denotes similarity and those with different letters are significantly different at $p < 0.5$.

The total quantities of fibre decomposition were calculated by accounting for biomass losses, which provides a clearer indication of the types of substrates being degraded. The total non-fibre and hemicellulose which was degraded showed a significant increase on day 21 in the less saturated microcosms and earlier on day 14 in the more saturated microcosms. Hemicellulose degradation would enable lignin degradation to proceed more rapidly due to enzyme accessibility to lignin as the close association between hemicellulose and lignin is separated [46]. Consequently, there appeared to be a correlation between hemicellulose and lignin degradation. Decomposition of the non-fibre was assumed due to growth on arabinoxylans which are present in a considerable proportion in wheat bran [21] and on hemicellulose. It was apparent that no cellulose was degraded by *T. versicolor* CM13, in contrast to higher cellulose degradation by the same fungus compared with hemicellulose degradation when grown on *Fraxinus excelsior* and *Acer pseudoplatanus* separately [7]. This may be due to the higher cellulose content in both woods between 40 and 45%, compared with 7% in wheat bran, while hemicellulose contents in these lignocellulose substrates were similar at 22%.

## 4. Conclusions

There are a number of factors that currently limit that the potential for commercial upscaling of SSF processes, including the type and particle of the growth substrate used, heat dispersion, aeration and provision of sufficient water to support growth and metabolic activity of the organisms under investigation.

This study reported the use of wheat bran, an agricultural by-product, as a growth substrate for the cultivation of *T. versicolor* CM13, a white rot fungus which produces lignin-degrading enzymes and the influence of moisture content on growth of this organism using SSF. The growth of *T. versicolor* CM13 on wheat bran revealed that the composition in terms of hemicellulose content and biomass remaining were different during active fungal growth but became similar during the stationary phase. In general, lignin-degrading enzyme activities were unaffected by different moisture conditions and were similar throughout the active and stationary phases. These results provide an insight into how minor differences in moisture content, which may be difficult to control, may affect SSF. This study revealed that the production of lignin-degrading enzymes using SSF could be achieved without moisture content being a limiting factor affecting fungal growth during colonisation.

**Author Contributions:** P.W.B., A.C.: Conceptualization, P.W.B.: investigation, P.W.B.: writing—original draft preparation, A.C.: writing—review and editing, A.C.: funding acquisition. All authors have read and agreed to the published version of the manuscript.

**Funding:** European Union's Interreg North West Europe program, financed by the European Regional Development Fund (Biowill: project number 964) and the Welsh Government (Environmental Evidence Program) for support.

**Institutional Review Board Statement:** Not applicable.

**Informed Consent Statement:** Not applicable.

**Data Availability Statement:** Available on request.

**Conflicts of Interest:** The authors declare no conflict of interest.

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
