# Peer review of "Establishing Experimental Conditions to Produce Lignin-Degrading Enzymes on Wheat Bran by Trametes versicolor CM13 Using Solid State Fermentation"

_waste, doi:10.3390/waste1030042_

Round 1

Reviewer 1 Report

In this manuscript, the authors have examined the solid-state fermentation of wheat bran using Trametes versicolor, grown under two moisture conditions, to determine conditions suitable for the production of lignocellulose-modifying enzymes. The authors determined that the higher moisture content conditions resulted in faster growth and increased hemicellulose degradation. Although lignin-modifying enzymes were not significantly different, the decomposition rates were distinct. Overall, this is a competently performed and clearly described study. I have only minor suggestions for improvement, which are detailed below.

General comment: review is greatly facilitated by line numbers. Since the authors used the journal-provided template, I’m not sure why there are absent. Please include line numbers in any revision.

Abstract

1.      “closely but different”? Similar but different, or distinct? Please clarify.

Introduction

2.      “Agricomycetes” – I believe the authors refer to Agaricomycetes - a Class of Basidiomycete fungi. Please correct.

3.      “genes and the expression of each of these enzymes” - genes are expressed, not enzymes/proteins. I think authors are referring to the expression patterns of the genes. Please clarify/correct.

Materials and Methods

4.      Section 2.2 – “were adjusted *with, or using* deionized water”  - please correct.

5.      Section 2.2 - how were the aerial mycelia prepared? Was a mycelial suspension blended? Please elaborate on inoculum preparation for the benefit of reproducibility.

6.      Section 2.3 - what numbers of replicates were performed at each of the moisture conditions? There clearly were replicates, as statistical analysis was performed. Please elaborate and clarify this point.

7.      Section 2.4 – “blended ()” did the authors mean to indicate the blending conditions? Is that what the () are for? Please clarify or correct.

8.      Section 2.4 – “molar *extinction* coefficient of ABTS…” Please correct.

9.      Section 2.4 – “50 hydrogen peroxide” – please provide units – presumably uM, but please clarify.

10.  Section 2.4- MnP assay – Did the authors consider using a boiled sample control, or a no peroxide control? This may have been better to control for any changes to A610 inherent to the sample. This is suggested below in the Remazol brilliant blue decolorization assay where the control had H2O2 withheld. Why was water used here and sample controls used for the other assays? Please explain the use of these controls for the respective assays.

11.  Section 2.4 – “being measured at 1 μM of the per min” – please clarify.

Results

12.  Since there is no discussion section, should this section be re-named, “Results and Discussion”? I do not know if the journal allows this format. If not, a Discussion must be added.

13.  Section 3.1 – “lignin content was significantly higher” - where did the "lignin" come from? This could be an artifact of the gravimetric determination of lignin content on digestion with 72% H2SO4 (section 2.4) - the weight of undigested fungal mycelia could add to the apparent lignin values. Please comment.

14.  Section 3.1 – Similarly, could fungal mycelia have contributed to this increase in ash content? Please comment.

15.  Tables 3,4 – Please explain the meaning of the lowercase letters (indicating statistically significant differences) along with the statistical test and p-values used.

minor writing issues detected; nothing that cannot be fixed in a few minutes

Author Response

  1. “closely but different”? Similar but different, or distinct? Please clarify.

Introduction

Has been changed to “two slightly different”

  1. “Agricomycetes” – I believe the authors refer to Agaricomycetes - a Class of Basidiomycete fungi. Please correct.

Has been corrected

  1. “genes and the expression of each of these enzymes” - genes are expressed, not enzymes/proteins. I think authors are referring to the expression patterns of the genes. Please clarify/correct.

The original sentence was replaced with “The combinations of the lignin degrading genes can vary significantly between different fungal species where for example, T. versicolor was shown to possess 12 short manganese peroxidase genes and 10 lignin peroxidase genes in the whole genome [10]”. 

Materials and Methods

  1. Section 2.2 – “were adjusted *with, or using* deionized water” - please correct.

Has been corrected

  1. Section 2.2 - how were the aerial mycelia prepared? Was a mycelial suspension blended? Please elaborate on inoculum preparation for the benefit of reproducibility.

The aim was to collect spores rather than aerial mycelium although the possibility that some aerial mycelium might be present has to be considered.  Additional details have been added as follows: “Sterile deionized water (10 ml) was added to the plates and the surface was scrapped aseptically with a loop.  The spore suspension was pipetted from the plates, into 30 ml sterile deionized water and mixed thoroughly.  Microscopy revealed a spore population of about 1x107 cells per ml with only a few fragments of aerial mycelium being present”.

  1. Section 2.3 - what numbers of replicates were performed at each of the moisture conditions? There clearly were replicates, as statistical analysis was performed. Please elaborate and clarify this point.

Triplicate microcosms were sampled and this has been included

  1. Section 2.4 – “blended ()” did the authors mean to indicate the blending conditions? Is that what the () are for? Please clarify or correct.

1 L Waring blender has been included within the brackets

  1. Section 2.4 – “molar *extinction* coefficient of ABTS…” Please correct.

“extinction” has been added

  1. Section 2.4 – “50 hydrogen peroxide” – please provide units – presumably uM, but please clarify.

Has been deleted because appears later on

  1. Section 2.4- MnP assay – Did the authors consider using a boiled sample control, or a no peroxide control? This may have been better to control for any changes to A610 inherent to the sample. This is suggested below in the Remazol brilliant blue decolorization assay where the control had H2O2 withheld. Why was water used here and sample controls used for the other assays? Please explain the use of these controls for the respective assays.

Boling causes precipitation of the proteins and results in lower absorbed interference. Consequently, this was determined to be an unsuitable control.  A sentence has been included to describe why samples were used without hydrogen peroxide as follows: “The sample changed the absorbance value of the assay and consequently it was necessary to include the sample within the control”.

  1. Section 2.4 – “being measured at 1 μM of the per min” – please clarify.

This has been deleted

Results

  1. Since there is no discussion section, should this section be re-named, “Results and Discussion”? I do not know if the journal allows this format. If not, a Discussion must be added.

The section has been renamed as Results & Discussion

  1. Section 3.1 – “lignin content was significantly higher” - where did the "lignin" come from? This could be an artifact of the gravimetric determination of lignin content on digestion with 72% H2SO4 (section 2.4) - the weight of undigested fungal mycelia could add to the apparent lignin values. Please comment.

An additional sentence has been included as follows: Klason lignin values were relatively high, and this may be due to the presence of cutin [32]. 

  1. Section 3.1 – Similarly, could fungal mycelia have contributed to this increase in ash content? Please comment.

Ashing was performed using standard NREL procedure although it is possible that lower ash contents might have been obtained if longer times had been used.  The total quantity of ash is more likely to remain constant whereas organic material will decrease due to decomposition.  Another sentence has been included as follows “The increase in ash content results from decomposition of organic material, rather than inorganic components which remain intact during fungal growth”.

  1. Tables 3,4 – Please explain the meaning of the lowercase letters (indicating statistically significant differences) along with the statistical test and p-values used.

A section has been included in the materials and methods as follows:

Statistical Analysis

Growth studies of T. versicolor CM13 on wheat bran over time and under different moisture conditions were conducted using duplicate and triplicate microcosms, respectively, and single samples were analysed from each microcosm.    Data analysis was conducted using ANOVA (analysis of variance) in SPSS version 27 with Tukey’s posthoc test.  The averages and standard deviations are shown in the figures and tables, while superscript letters represent significant differences.  Those with the same letters are similar and those with different letters are significantly different at P<0.05.

Reviewer 2 Report

waste-2513555

Title - Establishing Conditions to Produce Lignin Degrading Enzymes on Wheat Bran by Trametes versicolor CM13 Using Solid State Fermentation.

The manuscript by Baker and Charlton demonstrated a strategy for producing lignin-degrading enzymes under SSF using wheat bran by Trametes versicolor CM13. Overall, the manuscript is remarkable and requires major revision before its publication as follows:

Comments:

1.      Abstract, please add some quantitative results data on enzyme production activity.

2.      The authors should follow the significant number rule to present data in the text, and Tables.

3.      Page 2, “White rot fungi produce laccase,…………..these enzymes too [6].” Please add a few bacterial laccase sources and their lignin degradation and dye decolorizations application, i.e., https://doi.org/10.3389/fbioe.2019.00209.

4.      Please add this kind of information “The most complex component of lignocellulose is lignin, which is a hydrophobic heteropolymer composed of three major phenylpropane units: p‐hydroxyphenyl (H), guaiacyl (G), and syringyl (S).” i.e., https://doi.org/10.1002/biot.201800468.

5.      The discussion section is weak and requires significant updates to justify an in-depth explanation of findings. 

6.      Please add one illustration on the mechanism of lignin degradation/summary of the present study to highlight the significance.

Minor editing of English language is required

Author Response

1.  Abstract, please add some quantitative results data on enzyme production activity.

Variation of measured parameters as a percentage of actual values measured ranged from 1.33% to 144% in the unmixed microcosms and from 0.77% to 36.0% in the pre-mixed microcosms. 

Enzyme activities on day 21 in the microcosms at moisture content of 42.8% and at 54.6% for laccase activities were 750 ± 30.5 and 820 ± 30.8 units and for manganese peroxidase activities were 23.3 ± 6.45 and 21.4 ± 21.4 units, respectively. 

2. The authors should follow the significant number rule to present data in the text, and Tables.

Never heard of the significant number rule but have changed the data in the tables accordingly.  The text has been kept to one decimal place to help the reader.

3. Page 2, “White rot fungi produce laccase,…………..these enzymes too [6].” Please add a few bacterial laccase sources and their lignin degradation and dye decolorizations application, i.e., https://doi.org/10.3389/fbioe.2019.00209.

Reference 6 has described bacterial lignin degrading enzymes and has information from the reference has been expanded further as follows: e.g., Pseudomonas sp. and Cupriavidus basilensis have also been shown to possess peroxidases and laccases resulting in the production of lipids and polyhydroxyalkanoate.

4. Please add this kind of information “The most complex component of lignocellulose is lignin, which is a hydrophobic heteropolymer composed of three major phenylpropane units: p‐hydroxyphenyl (H), guaiacyl (G), and syringyl (S).” i.e., https://doi.org/10.1002/biot.201800468.

Another sentence has been included: Lignin is a major component within woods and grasses and is a highly crosslinked, large molecular weight polymer, composed of three different monolignols (p–coumaryl alcohol, guaiacyl alcohol, and sinapyl alcohol) and polyphenolic compounds [9]. 

Ponnusamy, V. K., Nguyen, D. D., Dharmaraja, J., Shobana, S., Banu, J. R., Saratale, R. G., ... & Kumar, G. (2019). A review on lignin structure, pretreatments, fermentation reactions and biorefinery potential. Bioresource technology271, 462-472.

5. The discussion section is weak and requires significant updates to justify an in-depth explanation of findings. 

 More has been written in the results & discussion section as shown by the red text and some sections have been rearranged.

6. Please add one illustration on the mechanism of lignin degradation/summary of the present study to highlight the significance.

The focus of this paper was to investigate the potential of this fungi to assist with lignin degradation and at this stage elucidation of the mechanism for this was not part of the scope of  work. . An additional and significant programme of research would be required before any mechanistic explanation could be proposed.

Reviewer 3 Report

The present ms is original and of interest to the readers. Major: The conclusion must contain a clear and founded reflection of the most important finding(s), i.e. laccase expression, based on chronological and moisture data. The methods must be more detailed and clearly relate to the results obtained. Various minr comments are included in the annotated ms.

Author Response

Major: The conclusion must contain a clear and founded reflection of the most important finding(s), i.e. laccase expression, based on chronological and moisture data. The methods must be more detailed and clearly relate to the results obtained. Various minr comments are included in the annotated ms.

The conclusions have been rewritten as follows:

There are a number of factors that currently limit that the potential for commercial upscaling of SSF processes, including the type and particle of the growth substrate used, heat dispersion, aeration and provision of sufficient water to support growth and metabolic activity of the organisms under investigation.

This study reports the use of wheat bran, an agricultural by-product, as a growth substrate for the cultivation of T. versicolor CM13, a white rot fungus which produces lignin degrading enzymes, and the influence of moisture content on growth of this organism using SSF.  The growth of T. versicolor CM13 on wheat bran revealed that the composition in terms of hemicellulose content and biomass remaining were different during active fungal growth but became similar during the stationary phase.  In general, lignin degrading enzyme activities were unaffected by different moisture conditions and were similar throughout the active and stationary phases.  These results provide an insight into how minor differences in moisture content, which may be difficult to control, may affect SSF.  This study reveals that the production of lignin degrading enzymes using SSF could be achieved without moisture content being a limiting factor affecting fungal growth during colonization.

Manufacturer/country -  Included Ankom, USA

more detail for method for insoluble lignin

The insoluble lignin was determined on the bags after ADF extraction using 500 ml of 72% (v/v) sulphuric acid in the Daisy machine (Ankom, USA) by rotating at room temperature for 3 h.  The bags were washed repeatedly in 2 L tap water until pH 7 which was measured using a pH meter.  () after blended must contain equipment and other information (1 L Waring blender)

Make of spectrophotometer, cuvette size - BioTek Epoch, Agilent, USA has been included

cm of what? Has been deleted

describe method for fungal biomass determination.

Has been deleted

place figure later in the paragraph

Has been moved

Klason lignin has not been mentioned in the methods section

This has been written as follows: Klason lignin contents were determined on a dried sample (0.3 g) using 72% sulphuric acid (5 ml) which was incubated at 25°C for 3 h and was regularly stirred.  Deionized water (240 ml) was added to the suspension, which was then autoclaved (121°C, for 1 h).  The hot suspensions were filtered through pre-weighed glass microfibre filters (Whatman) along with additional hot deionized water to remove any remaining soluble compounds.  The filters were oven dried (103°C) overnight and weighed and the lignin content was determined by subtracting the weight of the filter.  The ash contents were subtracted from these values.

Round 2

Reviewer 2 Report

Accept 

Reviewer 3 Report

I recommend publication of the study in its present form.